

# Circulation and oxygenation of the glacial South China Sea

**Shuh-Ji Kao[1], Tzu-Ling Chiang[2], Da-Wei Li[1], Yi-Chia Hsin[3], Li-Wei Zheng[1], Jin-Yu Terence Yang[1], Shih-Chieh Hsu[3], Chau-Ron Wu[2] and Minhan Dai[1]**

[1]State Key Laboratory of Marine Environmental Science, Xiamen University, Xiamen 361102, China

[2]Department of Earth Sciences, National Taiwan Normal University, Taipei, Taiwan

[3]Research Center for Environmental Changes, Academia Sinica, Taipei, Taiwan

*Correspondence to*: Shuh-Ji Kao (sjkao@xmu.edu.cn) and Da-Wei Li (ldw647@xmu.edu.cn)

**Abstract.** Degree of oxygenation in intermediate water modulates the downward transferring efficiency of primary productivity (PP) from surface water to deep water for carbon sequestration, consequently, the storage of nutrients versus the delivery and sedimentary burial fluxes of organic matter and associated biomarkers. To better decipher the PP history of the South China Sea (SCS), appreciation about the glacial-interglacial variation of the Luzon Strait (LS) throughflow, which determines the mean residence time and oxygenation of water mass in the SCS interior, is required. Based on a well-established physical model, we conducted a 3-D modeling exercise to quantify the effects of sea level drop and monsoon wind intensity on glacial circulation pattern, thus, to evaluate effects of productivity and circulation-induced oxygenation on the burial of organic matter. Under modern climatology wind conditions, a 135 m sea level drop results in a greater basin closeness and a ~23% of reduction in the LS intermediate westward throughflow, consequently, an increase in the mean water residence time (from 19 to 23 year). However, when the wind intensity was doubled during glacial low, the throughflow restored largely to reach a similar residence time (18.4 years) as today



regardless its closeness. Comparing with present day SCS, surface circulation pattern in glacial model exhibits (1) stronger upwellings at the west off Luzon Island and the east off Vietnam, and (2) an intensified southwestward jet current along the western boundary of the SCS basin. Superimposed hypothetically by stronger monsoon wind, the glacial SCS conditions facilitate greater primary productivity. Manganese, a redox sensitive indicator, in IMAGES core MD972142 at southeastern SCS revealed a relatively reducing environment in glacial periods. Considering the similarity in the mean water residence time between modern and glacial cases, the reducing environment of the glacial southeastern SCS was thus ascribed to a productivity-induced rather than ventilation-induced consequence.

## 1    Introduction

The South China Sea (SCS), located in the tropical-to-subtropical western North Pacific, is one of the largest marginal seas in the world. SCS acts as a connector between the Western Pacific Warm Pool and the East Asian monsoon system—these two engines drive the climate over East Asia continent (Wang, 1999). The monsoon-dominated seasonal patterns in bio-productivity and nutrient dynamics distinguish the SCS from other low-latitude waters that are insensitive to seasonal cycles (Wong et al., 2007b; Zhao et al., 2009a, and references therein); such features drew researchers' interests of its biogeochemical history and corresponding climate changes. Sedimentary records from the SCS, thus, have been oft-used to decipher the regional and global paleo-climate system (e.g., Chen and Huang, 1998; Chen et al., 2003; He et al., 2013; Jian et al., 2000; Kienast et al., 2001; Li et al., 2013; Li et al., 2011; Steinke et al., 2001; Wang et al., 2014; Wei, G. et al., 2003).

The modern surface circulation and the biogeochemistry in the SCS change drastically in response to the seasonal alternating East Asian monsoons (Gan et al., 2006; Liu et al., 2002; Shaw and Chao, 1994). In summer, the southwest monsoon drives an anti-cyclonic gyre (dashed curve in Fig. 1) in the southern



part of the basin, while in winter the northeast monsoon forces a cyclonic gyre covering the entire deep basin with an intensified southward jet along the east coast of Vietnam (Fang et al., 1998; Liu et al., 2002; Wyrtki, 1961). The summer monsoon excites upwelling thus leading to higher primary productivity off the east coast of Vietnam, while the winter monsoon triggers upwelling northwest off

Luzon and north of the Sunda Shelf (Liu et al., 2002) and a stronger basin wide diapycnal mixing.

On the other hand, a fairly short residence time of 30–120 years was suggested for the water mass in the SCS basin scale (Broecker et al., 1986; Gong et al., 1992; Qu et al., 2006). The Luzon Strait (LS) between Taiwan Island and the Philippines is the principal deep channel (sill depth ~2,400 m) allowing surface and deep water exchange between the SCS and the western Pacific Ocean (Fig. 1), whereas

other channels are only effective at high sea level, such as Taiwan Strait (~70 m), Karimata Strait (~50 m) and Balabac Strait (~100 m), except the Mindoro Strait (~420 m) being sufficiently deep for water exchange at low sea level stand (Wang and Li, 2009, and references therein). The deep water flows from the Philippine Sea into the SCS over the sill at ~2100 m deep, then upwells and mixes with surface waters to form the intermediate water (Gong et al., 1992), which flows out of the LS exhibiting a

"sandwich-like" flow pattern (Chao et al., 1996; Li and Qu, 2006; Qu et al., 2006; Tian et al., 2006).

According to the circulation pattern and geomorphologic closure with deep entrance at the northeast (Fig. 1), modern day observational data displays a significant oxygen supply from the LS and an oxygen minimum zone at 500–1500 m water depth at the southern and southeastern parts of the SCS (Qu, 2002; Li and Qu, 2006). During glacial times, the oxygen level in the SCS intermediate water is likely less,

particularly at the southern basin, as inferred from benthic [13]C values, benthic foraminiferal assemblages and geochemical evidences collected at bottom depth of ~1600 m (Jian et al., 1999; Wang et al., 1999). On the other hand, the contents of sedimentary organic carbon and redox sensitive elements in sediment core MD972142 retrieved from the southeastern SCS (at 1557 m) reveals a reducing state during glacial time (Löwemark et al., 2009).



Meanwhile, a consensus had been reached in general that a stronger winter monsoon plays a major control on primary productivity (PP) for glacial periods (Chen et al., 2003; He et al., 2013; Li et al., 2014) although some reports displayed opposite results off the east coast of Vietnam (Wei, G. et al., 2003). Combined with the circulation pattern aforementioned, the deep water in the SCS was speculated

to be less oxygenated due to increased PP and restricted circulation caused by sea level drop (Wang et al., 1999), particularly, the southeastern part of the basin (Löwemark et al., 2009). Although paleo-environmental proxies were strongly associated with the dynamic changes in sea level caused by the waxing and waning of the continental ice sheets, unfortunately, no modeling work has been done to simulate the physics of the circulation pattern and basin wide ventilation in terms of paleoceanographic

perspective.

In this paper, we use a well-established 3-D ocean model to investigate how the circulation pattern in the SCS responds to changes in monsoon intensity and sea level induced geomorphic configuration. Model output displays a slightly weakened throughflow when sea level drops to the last glacial condition; yet, intensified winter monsoon wind exerts an opposite effect for compensation. Our

modeling work provides physical oceanographic evidences benefiting future studies on deciphering the paleo-environmental proxies recorded in the SCS.

## 2   Material and methods

### 2.1   Numerical model

The East Asian Marginal Seas (EAMS) model (Hsin et al., 2008, 2010, 2012; Kao et al., 2006; Wu et al., 2008; Wu and Hsin, 2005), based on the Princeton Ocean Model (Mellor, 2004) with realistic topography, is used in this study. The EAMS model covers a domain of 99°E–140°E and 0°N–42°N with a resolution of $1/8° \times 1/8°$ in horizontal and has 26 sigma levels in vertical. On the open





boundaries, the EAMS model derives its boundary conditions from a larger-scale North Pacific Ocean (NPO) model. The NPO model domain covers the entire Northern Pacific ranging from 99 °E to 71 °W in longitude, and from 30 °S to 65 °N in latitude with a horizontal resolution of $1/4° \times 1/4°$. Its predictive capability in simulating 3-D seasonal circulations in the studied domain had been validated by a log-term data of temperature and salinity observed in the northern SCS (Tseng et al., 2005), current velocity data from several mooring stations in the SCS and satellite altimetry (Hsin et al., 2012). These validations and more detailed descriptions of the SCS model are given in Wu and Chiang (2007), Chiang et al. (2008) and Hsin et al. (2012). Overall, the SCS model simulate very well in term of transport, flow path and vertical distributions of temperature and salinity.

The EAMS model was forced by 10-year (1999–2008) repeated wind data from 6-hourly QuikSCAT/NCEP blend (http://rda.ucar.edu/datasets/ds744.4/) ocean surface wind datasets and 10-year (1999–2008) repeated heat flux from daily Modern Era Retrospective-analysis for Research and Applications (MERRA, http://gmao.gsfc.nasa.gov/research/merra) reanalysis datasets with 10-year repeated boundary conditions from daily NPO model.

Four experiments are carried out for this study. Every case was run for 60 years, and the results in the last 10 years are used for the present analysis. During the LGM, the sea level was about 135 m lower than today (Wang, 1999, and references therein). Thus we set the glacial case at sea level of -135 m (the lowest sea level, LSL) and modern day case at 0 m (the present sea level, PSL).

To examine changes in circulation pattern during different climate state, both the PSL and LSL cases were conducted (Table 1). We also doubled the climatological wind intensities for comparison, therefore, total 4 cases were conducted (Table 2). The modeled surface circulation pattern was presented by integrating the upper 100 m flow. Since the LS is the primary entrance, we presented its cross-sectional transports as well as the integrated vertical transport profiles for all cases as example in annual integration basis. More details, such as validation of the model and process of elimination




experiments to assess the relative importance of open ocean inflow/outflow, wind stress, and surface heat flux, in regulating the LS transport and its seasonality were presented in Hsin et al. (2012).

## 2.2 Sample and chemical analysis

Core MD972142 (12°41.33'N, 119°27.90'E, water depth: 1557 m) was retrieved from the northwest off Palawan Island during the IMAGES III-IPHIS cruise in 1997 (Chen et al., 1998). Detailed sediment property was described by Chen et al. (1998), and age model is based on AMS $^{14}$C data and planktonic foraminiferal $\delta^{18}$O correlation (Wei, K. Y. et al., 2003, and references therein). A total of 129 samples were taken for metal elements analysis, mostly at 20 cm intervals, for the past 502 kyr. For metal analysis, about 0.2 g dry sample was digested using an acid mixture of HF, HNO$_3$ and HClO$_4$ according to the procedure described by Hsu et al. (2003). The digested solution, which contains dissolved metals were analyzed for Mn, Ti and Al using an ICP-OES (Optima 3200DV, Perkin-Elmer$^{TM}$ Instruments, USA).

Mn is redox-sensitive element with different valence states. Higher oxidation states of Mn (III) and (IV) occurring as insoluble oxyhydroxides, and are found in well-oxygenated environments; while under oxygen-depleted settings, these Mn oxides can be reduced to Mn (II) which is more soluble and mobile (Calvert and Pedersen, 1996). Thus, the presence of excess Mn suggests an oxygenated environment. Considering sedimentary Al is only terrestrial origin and it is not affected by biological or diagenetic processes (Brumsack, 2006), we generate Mn/Al ratio (enrichment factor) to reconstruct the redox state history. In fact, Mn content in core MD972142 was determined by XRF (X-ray fluorescence spectrometer) and had been reported by Löwemark et al. (2009); however, our ICP-OES provide more quantitative result. Total organic carbon (TOC) reported by Chen et al. (2003), C$_{37}$ Alkenones by Shiau et al. (2008) and planktonic foraminiferal $\delta^{18}$O by Wei, K. Y. et al. (2003) were drew for discussion.



## 3 Results and discussions

### 3.1 Surface circulation patterns

Integrated flow patterns of upper 100 m in annual average basis were presented in Figure 2. The flow patterns were analogous in four cases. However, three major features in the flow pattern can be identified. Firstly, a clockwise loop pattern appeared in all cases around the LS. This loop pattern indicates a fast exchange in the surface 100 m that most of the water coming into the SCS veering out in a short time regardless distinctive intensities among cases. The loop exchange in LSL cases were apparently less. Secondly, two cyclonic eddies; one to the west off Philippine Island and the other to the east off Vietnam can be seen. In modern day, the former one occurs in winter time when northeasterly monsoon is intensified and Kuroshio intrusion is enhanced and the latter one appears in summer time when southwesterly wind prevails (Qu, 2000; Su, 2004). Both areas have been identified as upwelling zones in modern day. Also, we can see clear differences in eddy intensity. The lowest intensity for both eddies took place in two cases with standard wind while the most intensified eddies appeared in the two cases with doubled wind regardless the sea level change. Wind intensity is an obvious driver for these two eddies. The third feature was the southwestward jet flow along the western boundary. This southwestward jet flow was intensified particularly in high wind cases too. As indicated by magnetic properties and grain size spectrums in cores from the northern SCS (Zheng X. F., personal communication), intensified west boundary current in glacial time is highly likely. In general, the model catches main features of the SCS modern upper water circulations.

The most important message conveyed by such comparison is that wind governs the surface flow pattern. When comparing with the PSL case with normal wind (Figure 2a), the stronger wind in glacial period (Figure 2d) may not only promote the vertical mixing to bring up nutrients from subsurface but also enhance the two eddies associated with upwelling to further fuel the primary productivity. By using the foraminifera-bound $\delta^{15}N$ in the core MD972142, Ren et al. (2012) had proved a shoaling nitracline



during the sea level low in the last glacial cycle. In addition, reconstructions of upper ocean thermal gradient revealed a deepened mixed layer throughout the SCS basin (i.e., from northern ODP site 1147 to southern core MD01-2390) during glacial period (Li et al., 2013; Steinke et al., 2010). In summary, paleo-records support our model results that stronger monsoon wind promotes upper water mixing during sea level low.

## 3.2   Sea level and wind forcing transport in the SCS

Table 2 reveals that LS is the major entrance holding the largest inflows in all experimental cases regardless sea level and wind intensity changes. All straits serve as outlets except some in specific cases, such as annual and winter transports through Balabac Strait in PSL model runs, summer transport through Karimata Strait under PSL and summer transport through Mindoro Strait in PSL normal wind case. Under PSL condition, Taiwan Strait and Mindoro Strait play the most important role channeling water out among all straits, and Mindoro Strait contributed more in the case with doubled wind. While under LSL conditions, topographic configuration made Mindoro Strait to be the only exit for the coming water through LS. For each individual case, the mass balance was reached (summation of all positive and negative values in Table 2).

For the winter time of normal wind case under PSL condition (Win of QSK in Table 2), the net westward transport through LS is 7.53 Sv. Outflow is displayed by a northward flow to the East China Sea at a rate of 0.88 Sv through Taiwan Strait, by a net southeastward flow into the Sulu Sea at rate of 3.44 Sv through Mindoro Strait and Balabac Strait, and by southward flow into the Java Sea at a rate of 3.22 Sv through Karimata Strait. For the summer time in QSK case, net water inflows into the SCS basin are operated by West Philippine Sea through LS, as well as by Java Sea and Sulu Sea through relevant straits, while the water mass balance is maintained mainly by outflow into the East China Sea though the Taiwan Strait. The net inflow in summer (3.36 Sv) is about 41% of the winter time. Apparently, northeasterly monsoon wind is the key driver for water exchange.



Compared to PSL cases, cases under LSL exhibited 33%–60% reduction in integrated LS inflow under corresponding time period. Such significant reduction in LS throughflow highlights the topographic effect, subsequently, the mean residence time of water mass in the SCS. Model runs under LSL conditions (Table 2) again revealed the importance of wind intensity, e.g., the doubled wind cases (HQSK2) hold at least 54% higher exchange rates through LS (93% for annual, 54% for winter and 314% for summer) when comparing to normal wind cases (HQSK).

The cross sections of annual mean zonal velocity (m s$^{-1}$) along 120.8 °E for four experiments were displayed in Figure 3. In all four cases, water flows in and out at any depth exhibiting a major inflow core at the center of LS that can even extend to 1000–1500 m. While the most significant outflow core appears in the upper 500 m at the very north close to Taiwan Island. The flow pattern in upper 100 m is in accordance with the clockwise loop current described in part 3.1 and Figure 2, which has been identified by many modern surveys (Fang et al., 1998; Lan et al., 2004). Worthwhile to note that the intensities of loop current in upper 500 m in PSL cases were larger than those in LSL cases indicating the sea level height is beneficial to surface exchange.

As for the water interval of 500 m to 1200 m or so, the magnitude of integrated flow rate for inflow and outflow was indistinguishable; yet, this interval possessed more inflow (blue color), which differed from some of the recent short-term observations with outflowing in summer. In fact, the present LS transport has great spatio-temporal variability (Hsin et al., 2012) and the largest transport took place in winter time. Hsin et al. (2012) compiled all previously documented short-term current observations, among which only one observation was carried out in winter by Yuan et al. (2012). The inflow speed determined by their Argo floats at 1000 m is ~0.15 m s$^{-1}$, agreeing well with our model results at this specific depth. Since the Luzon transport is highly season- and site- dependent, short term observational data at limited sites may not be sufficient to represent the annual integration over spatial scale. More cross sectional longer term observations are needed to pin down the issue of throughflow transport.



### 3.3   LS intermediate inflow and mean water residence time

Annual mean net transports per unit depth (Sv m$^{-1}$) of LS transection for four experiments were shown in Figure 4, in which a clear "sandwiched structure" was revealed in all cases. Take the PSL and normal wind condition case (QSK, black curve in Figure 4) as example, we can see net outflows of the SCS water into the Philippine Sea at the surface (20–135 m) and in the deeper part (1190–2430 m) with a strong net westward transport at intermediate depth (135–1190 m) as well as a tiny inflow below 2430 m (Hsin et al., 2012). This four level "sandwiched structure" water exchange pattern has also been revealed by numerical model in Fang et al. (2009), though their depths of boundary layer differ from ours.

In PSL case under doubled wind condition (red curve in Figure 4), the surface outflow was turned into inflow and the inflow of intermediate layer (135–1190 m) was intensified. During LSL condition, this sandwiched vertical structure changed as follows for both normal and doubled wind cases: (1) both intensity and depth of net surface eastward transport increased (HQSK and HQSK2); (2) the upper boundary of the westward inflow migrates to deeper depth at ~310 m, with the largest inflow centering around 600–900 m.

Instead of the surface layer, which exchanges rapidly, the intermediate inflow through LS determines the basin-scale water residence time, subsequently, the magnitude of oxygen minimum zone of the SCS interior. By dividing the SCS volumes by corresponding intermediate inflow rate (depth-integration of negative values for intermediate layer) under PSL and LSL, we calculated the mean residence time for the water mass in the SCS on basin scale (Table 3). Note that here we set variable upper and lower boundaries, which were defined as starting and ending depths of negative values in transport per unit depth (see Fig. 4), for the intermediate inflow above 1500 m. For the normal wind speed and PSL condition (see QSK in Table 3), the residence time of 19.0 years is in agreement with previously published result by Qu et al. (2006), i.e. less than 30 years for modern SCS; but shorter than that



reported by Chen et al. (2001), i.e. 40–50 years. However, when sea level dropped to the LGM condition under normal wind speed (HQSK), the water residence time increased to 23.0 years (~20%) due to the reduction of intermediate inflow (Table 3). During low sea level stand, emerged Sunda Shelf and Taiwan Strait constrained the SCS exchange leaving the LS and the Mindoro Strait as the sole inlet

and outlet, respectively, thus leading to the reduction of deep water replenishing rate. However, the increase of residence time from 19 years to 23 years due to sea level drop is not as significant as anticipated since the total volume of SCS reduced concomitantly as the reduction of intermediate inflow.

The effect of seasonally reversing East Asian monsoon winds on the SCS basin scale water ventilation is evident also. In PSL doubled wind case, the intermediate transport increased significantly

as a consequence. Meanwhile, results demonstrate that winter season net westward through LS is ca. 3.8 times higher than summer season, and this is consistent with plenty of field observations (Lan et al., 2004), and numerical models carried out in the LS (Fang et al., 2005; Song, 2006; Wang et al., 2009; Yaremchuk et al., 2009; Zhao et al., 2009b). For the LSL condition with doubled wind speed enforcement (HQSK2), increased westward transport results in a residence time of 18.4 years, which is

nearly the same as that of the PSL condition.

During glacial, stronger East Asian winter monsoon (EAWM) has been indicated by plenty of evidences from the Chinese Loess Plateau using mean grain size (An, 2000; Porter and An, 1995; Sun et al., 2006), from the Chinese lake sediments using titanium content (Yancheva et al., 2007), and from the SCS using the difference of upper water temperatures (Li et al., 2013; Steinke et al., 2010). On the other

hand, paleo-records suggest glacial stronger EAWM was accompanied by weaker East Asian summer monsoon (EASM) (Wang et al., 2001; Yuan et al., 2004), which indicated our model case of doubled wind stresses for both winter and summer season may overestimate. However, annual mean transport incorporates more winter season signals since the volume transport in winter is 2–3 times higher than that in summer, and the increment induced by doubled winter wind are higher than that of doubled

summer wind.



Basing on our model results (Table 2, 3 and Figure 4), it is reasonable to hypothesize the water residence time of the glacial SCS is in between of 18.4–23.0 years, very close to that in present day (19 years). Previous report indicated that during the last glacial period, the radiocarbon age of SCS deep water (at depth of 2695 m) was estimated by using [14]C age differences between planktonic and benthic

foraminifera cells to be 1670±105 years (Broecker et al., 1990), which is almost identical to that of today (1600 years; Broecker et al., 1990) suggesting a similar ventilation during low sea level period (although the [14]C age uncertainty is larger than the water residence time). Our model results support above finding.

Recently, Wan et al. (2014) applied the same age-difference method but to infer the environmental

change (mainly vertical mixing) between northern and southern SCS for the past 30 kyr. Their results suggested that in the southern SCS  the vertical mixing and advection in the upper water column is more vigorous in the Holocene than in glacial. By contrast, our modern day case (i.e., Holocene) revealed weaker eddy intensity and southwestward jet flow for the southern SCS. The reason is not known, however, the fresh water input, a factor might be important in the southern SCS, was not considered in

our model. On the other hand, Wan et al. (2014) also found similar vertical mixings at the northern SCS basin between the Holocene and the glacial periods. This part agrees with our model results.

Note that there should be a wide range of residence time for different water parcels in various locations in the SCS basin. Moreover, we clearly knew that the water replenishment in winter time is stronger than that in summertime; thus, residence time is also season-specific. Here in this paper, the

residence time we presented was an estimate of mean annual state basing on the concept of box model. The potential way to probe the residence time for specific water parcel is to release tracers with given density range along the Luzon Strait for longer model runs. More model studies are needed in future to resolve this issue, particularly, the route and residence time of intermediate water, which may regulate the efficiency of export production and the inventory of nitrate (via denitrification), thus, the primary

production and subsequent burial of organic carbon for entire basin.



### 3.4 Oxygen in intermediate water

The westward intermediate inflow spreads over the SCS basin resulting in distinctive physical and chemical properties (Wang and Li, 2009; Wong et al., 2007a; Wong et al., 2007b). Higher oxygen contents at intermediate water depth always appear at the northern SCS (Li and Qu, 2006; Qu, 2002), and oxygen deficient zones occurred at around 550–1000 m toward the southern basin of the SCS (Li and Qu, 2006; Qu, 2002). Such spatial oxygen distribution pattern supports the lateral supply of oxygen from intermediate inflow through LS. Given that the oxygen utilization rate (OUR) of intermediate waters in ocean interior is primarily attributed to the decomposition of sinking particulate organic matter (POM), here we drew sinking POM data to estimate OUR for discussion and validation.

The recent reports of particulate organic carbon export from the euphotic zone (based on radionuclide tracers and floating trap) in the SCS basin ranged from 6.3 mmol C m$^{-2}$ day$^{-1}$ to 14.4 mmol C m$^{-2}$ day$^{-1}$ (Cai et al., 2015; Wei et al., 2011). Assuming that the decrease of particulate organic carbon fluxes during its downward transit in the SCS fits the Martin Curve (Martin et al., 1987), we may obtain the amount of decomposed particulate organic carbon at 300–1500 m depth interval (1.7–3.9 mmol C m$^{-2}$ day$^{-1}$). On the other hand, by direct sediment trap observation at ~300–500 m in the northeast basin the particulate carbon fluxes ranged from 1.6 mmol C m$^{-2}$ day$^{-1}$ to 6.9 mmol C m$^{-2}$ day$^{-1}$ with a time integrated mean of 4.0 mmol C m$^{-2}$ day$^{-1}$ (Kao et al., 2012). In the central SCS basin, the average sinking POM collected below 1500 m was ~0.3 mmol C m$^{-2}$ day$^{-1}$ (Gaye et al., 2009). Available sinking POM data is limited, however, sinking POM fluxes at the northern SCS basin are higher than those in the southern basin due to the remarkable influence of lateral transport from shelf and slope regions and upwelling areas where the primary production are high (Liu et al., 2007; Yang et al., in preparation).

Basing on above sinking POM estimates, the OUR in the SCS was calculated to be 0.7–1.8 µmol kg$^{-1}$ yr$^{-1}$ (by considering that the ratio of organic carbon to $O_2$ consumption during organic matter remineralization is 106:138). By using bomb $^{14}$C and/or chlorofluoro-carbon concentrations, OUR of





the North Pacific intermediate water (NPIW) was estimated to be 3.2 µmol kg$^{-1}$ yr$^{-1}$ to 4.6 µmol kg$^{-1}$ yr$^{-1}$ (Feely et al., 2004; Sonnerup et al., 1999), which is higher than the trap-derived values in the SCS. In fact, by assuming the oxygen utilization rate of 3 µmol kg$^{-1}$ yr$^{-1}$, You et al. (2005) derived a younger age of 9–14 years for the SCS intermediate water (originated from NPIW intrusion). If we set an OUR of 1.5 µmol kg$^{-1}$ yr$^{-1}$ for entire SCS basin, we may obtain a residence time of 18–28 years, which is more consistent with our modeled residence time.

Besides residence time, the initial oxygen content of the intermediate inflow also plays a role in the degree of basin oxygenation. During glacial condition, lower glacial-stage sea surface temperature may increase oxygen solubility in high-latitude regions, while stronger winds further enhance the rate of thermocline ventilation resulting in colder, rapidly flushed intermediate water, consequently (Galbraith et al., 2004). Accordingly, the NPIW should be characterized by relatively higher oxygen in glacial periods. By assuming both LS inflow intermediate water oxygen content (130 µmol kg$^{-1}$) and oxygen utilization rate (1.5 µmol kg$^{-1}$) are the same as today, the oxygen content of intermediate water at the southern SCS can be derived to be 95 µmol kg$^{-1}$ for the HQSK case and 102 µmol kg$^{-1}$ for the HQSK2 case (LSL cases). The two dissolved oxygen values are similar and sufficiently high not reaching the criterion for water column denitrification (Naqvi et al., 2006). Considering the assumption that oxygen content in LS inflow intermediate water during glacial might be higher than the present value, the low water oxygen content during glacial time (if any) can only be achieved by increasing the productivity once we set the glacial mean residence time the same as today.

## 3.5 Productivity-driven redox changes and organic matter preservation

Since the core MD972142 record reveals a mirror image between the low frequency temporal variation of TOC and sea level curve, Löwemark et al. (2009) attributed such inverse relation to restrict ventilation, thus, reducing water column to preserve organic carbon. The reducing environment might exist, however, not necessarily to be driven by ventilation or basin wide exchange according to our



model results. Various paleoceanographic records over the recent glacial cycles showed higher export production associated with stronger winter monsoon in the northern SCS basin (He et al., 2013; Higginson et al., 2003; Huang et al., 1997a; Huang et al., 1997b; Li et al., 2014; Tamburini et al., 2003). For the southeastern SCS, TOC and organic biomarkers indicate higher primary productivity during

glacial periods being ascribed to stronger EAWM (Li et al., 2014; Shiau et al., 2008). As aforementioned, our model case in glacial condition favors vigorous diapycnal mixing to support paleo-records of increased primary productivity. It has been proposed that large phytoplankon can transport carbon downward more efficiently (Boyd and Newton, 1999; Zhao et al., 2009a), and higher diatom productivity and relatively high diatom contribution to total PP have been reported during glacial in the

southeastern SCS (Li et al., 2014). Thus, higher fractions of upper euphotic photosynthesized organic carbon would escape the water column degradation and eventually buried in sediment, without altering the water column oxygen content significantly during glacials. In fact, during glacial periods globally reduced ocean subsurface water column denitrification had been proved (Galbraith et al., 2013). On the other hand, bulk sedimentary $\delta^{15}N$, which is used to track water column denitrification, widely observed

in the SCS had not changed throughout 200 kyr regardless the sea level fluctuations (Kienast, 2000). This invariable bulk sedimentary $\delta^{15}N$ suggested that oxygen content in the SCS water column remained similar during glacial-interglacial cycles. All evidences aforementioned suggested a similar oxygenation condition in water column during glacial period even though the surface production had increased.

Below we pulled geochemical data to discuss the sedimentary redox change induced by enhanced delivery of organic matter to the bottom sediments. The Mn/Al ratio from core MD972142 (Figure 5) displays clear glacial-interglacial variation during the past 502 kyr. Interglacial Mn/Al ratios are much higher than average shale value $9.62 \times 10^{-3}$ (Brumsack, 2006), suggesting an enrichment of Mn; whereas most glacial Mn/Al ratios are lower than that of average shale suggesting a remobilization of

Mn. In general, reducing or lower sedimentary oxidation state were required to maintain Mn deficiency.



Before applying the Mn/Al ratio we need to confirm sediment Al, which is often used to infer terrestrial input background (Brumsack, 2006), in our core was conservative not been influenced by biogenic process. Enrichment of Al had been reported (Wei, G. et al., 2003; Murray and Leinen, 1996) during biogenic opal formation. However, in our core MD972142 the opal record revealed high values during

interglacials (Shiau et al., 2008), whereas high Al values mostly appeared in glacials (Fig. 5A). On the other hand, temporal variation of Ti (Figure 5A), another element used in monitoring terrestrial input background, displayed an almost identical temporal pattern to Al and a good linear correlation ($r^2 = 0.67$; $p < 0.05$) between Ti and Al was observed. Above evidences suggested biogenic effect on sedimentary Al was negligible in our studied region. According to discussions above, the variability of Mn/Al ratio

should be mainly caused by redox-induced mobilization. In addition, total sulfur contents from core MD972142 (unpublished data) display peak values during glacials while low values during interglacials, supporting the scenario of glacial sedimentary suboxic condition. Interestingly, the temporal variation of Mn/Al ratio (Fig. 5B, analyzed using X-ray fluorescence spectrometer) previously reported by Löwemark et al. (2009) for the northern SCS (core GIK17925-3, 2980 m water depth) was less variable

during glacial/interglacial cycles partly because of its deeper water depth and partly because of its location near the LS entrance where oxygen level was constantly high due to vigorous exchanges around the LS entrance.

Since organic matter delivered to sediment surface contains high fraction of liable organics (Tyson, 1995), whether the variable organics preservation efficiency caused by sedimentary redox changes may

affect the applications of productivity proxies remains unknown; but if such case happened it should happened in the southeastern SCS rather than in the northern SCS, where oxygenation status was relatively constant. For instance, the reported temporal variation of $C_{37}$ Alkenones in core MD972142 (Fig. 5E), a productivity indicator of haptophytes (mainly produced by *Emiliania huxleyi* and *Gephyrocapsa oceanica*), largely followed the pattern of total organic carbon content (Fig. 5D)

revealing higher values during glacials (e.g., MIS 6, 8 and 10). Although our model results and



geochemical proxy records suggested that the reducing sedimentary environment in glacial southeastern SCS was ascribed to a productivity-induced rather than ventilation-induced consequence; it does not mean the sedimentary biomarkers may truthfully reflect the surface ocean productivity. Cautions should be made while reconstructing the SCS productivity history, particularly based upon organic biomarker methods in redox sensitive areas like the southeastern SCS.

## 4 Conclusions

Winter northeasterly monsoon wind intensity governs the volume transport of Kuroshio intrusion through the Luzon Strait, subsequently, the water exchange rate and the mean residence time of water body of the SCS. Although sea level drop promotes geomorphic closure resulting in less westward LS intermediate throughflow (6.96 Sv reduced to 5.33 Sv), the mean water residence time of SCS deep water did not increase proportionally due to a synchronous reduction in water body. By strengthening the monsoon wind, similar water residence time can be reached for glacial low sea level stand. Resembling residence time between glacial and present today eliminates the restriction ventilation hypothesis. However, to reach lower redox state observed in glacial time, higher productivity is required. Lacking of bulk $\delta^{15}$N variation rules out water column denitrification, thus, we suggest the reducing condition mainly appeared for sedimentary environment. The glacial-interglacial sedimentary redox state change may induce conflicts among interpretations by organic and inorganic proxies; meanwhile, the organic biomarkers may not truthfully reflect the surface productivity due to preservation efficiency change. More sophisticated observations and modeling studies, such as deploying Bio-Argo at multiple depths in deep water to monitoring biogeochemical evolution along the courses, using neutral buoyancy tracer in 3-D models to examine forward trajectory and asymmetrical monsoon wind enhancement, are needed to probe the flow path of intermediate water and to obtain comprehensive understandings multi-dimensionally and a better approximation of residence time for specific water parcel. Nevertheless, our

modeling exercise advances current knowledge on the possible surface circulation patterns and deep water ventilation of the South China Sea and offers hydrodynamic information to certain degree for the interpretations of biogeochemical and sedimentological clues retrieved from sediment cores.

## Acknowledgements

This study was supported by National Natural Science Foundation of China (NSF, Grant No. 91328207, and 41176059), and by the National Basic Research Program of China (973 Program, Grant No. 2015CB954003).

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



Table 1. The environmental settings for model simulation.

| Case label | PSL[a] | PSL | LSL[b] | LSL |
| | QSK[c] | QSK2[d] | H[e]QSK | HQSK2 |
| --- | --- | --- | --- | --- |
| Sea level | 0 m | 0 m | -135 m | -135 m |
| Wind type | Normal | Double | Normal | Double |

[a]PSL: present sea level. [b]LSL: lowest sea level. [c]QSK: Wind forcing by QuickSCAT/NCEP blend (1999–2008). [d]2: Doubling the wind magnitude. [e]H: Setting sea level at -135 m.



Table 2. The model results of net transport of five straits.

| Case label | QSK | | | QSK2 | | | HQSK | | | HQSK2 | | |
|---|---|---|---|---|---|---|---|---|---|---|---|---|
| | Ann[a] | Win[b] | Sum[c] | Ann | Win | Sum | Ann | Win | Sum | Ann | Win | Sum |
| LS[d] | -3.63 | -7.53 | -0.84 | -6.85 | -12.56 | -2.59 | -1.64 | -3.28 | -0.42 | -3.16 | -5.05 | -1.74 |
| TS[e] | 1.92 | 0.88 | 3.21 | 0.96 | -0.70 | 3.25 | - | - | - | - | - | - |
| KS[f] | 0.80 | 3.22 | -1.83 | 1.33 | 4.95 | -2.47 | - | - | - | - | - | - |
| BS[g] | -0.23 | -0.65 | 0.15 | -0.05 | -0.54 | 0.32 | - | - | - | - | - | - |
| MS[h] | 1.13 | 4.09 | -0.69 | 4.62 | 8.84 | 1.52 | 1.64 | 3.28 | 0.42 | 3.16 | 5.04 | 1.75 |

Note: Unit in Sv. Positive number means transport out from the SCS; Negative number means transport into the SCS; "-" mean no data due to the strait outcrops during low sea level conditions; [a]Ann: Last 10 years mean; [b]Win: Last 10-year Dec-Feb mean; [c]Sum: Last 10-year Jun-Aug mean; [d]LS: Luzon Strait;

5   [e]TS: Taiwan Strait; [f]KS: Karimata Strait; [g]BS: Balabac Strait; [h]MS: Mindoro Strait.





Table 3. Estimated residence time for the SCS based on the net transport of intermediate water through the Luzon Strait.

| Case lable | QSK | QSK2 | HQSK | HQSK2 |
|---|---|---|---|---|
| LS TP interval (m) | 135–1190 | 20–1190 | 310–1230 | 310–1270 |
| TP (Sv) | -7.04 | -10.41 | -5.35 | -6.71 |
| RT (year) | 19.0 | 12.8 | 23.0 | 18.4 |

Note: Westward transport (TP) interval through Luzon Strait is based on Figure 4. Negative number means transport into the SCS; Water volume used for calculation is ~$4.21 \times 10^{15}$ m$^3$ and ~$3.88 \times 10^{15}$ m$^3$ of PSL and LSL, respectively.



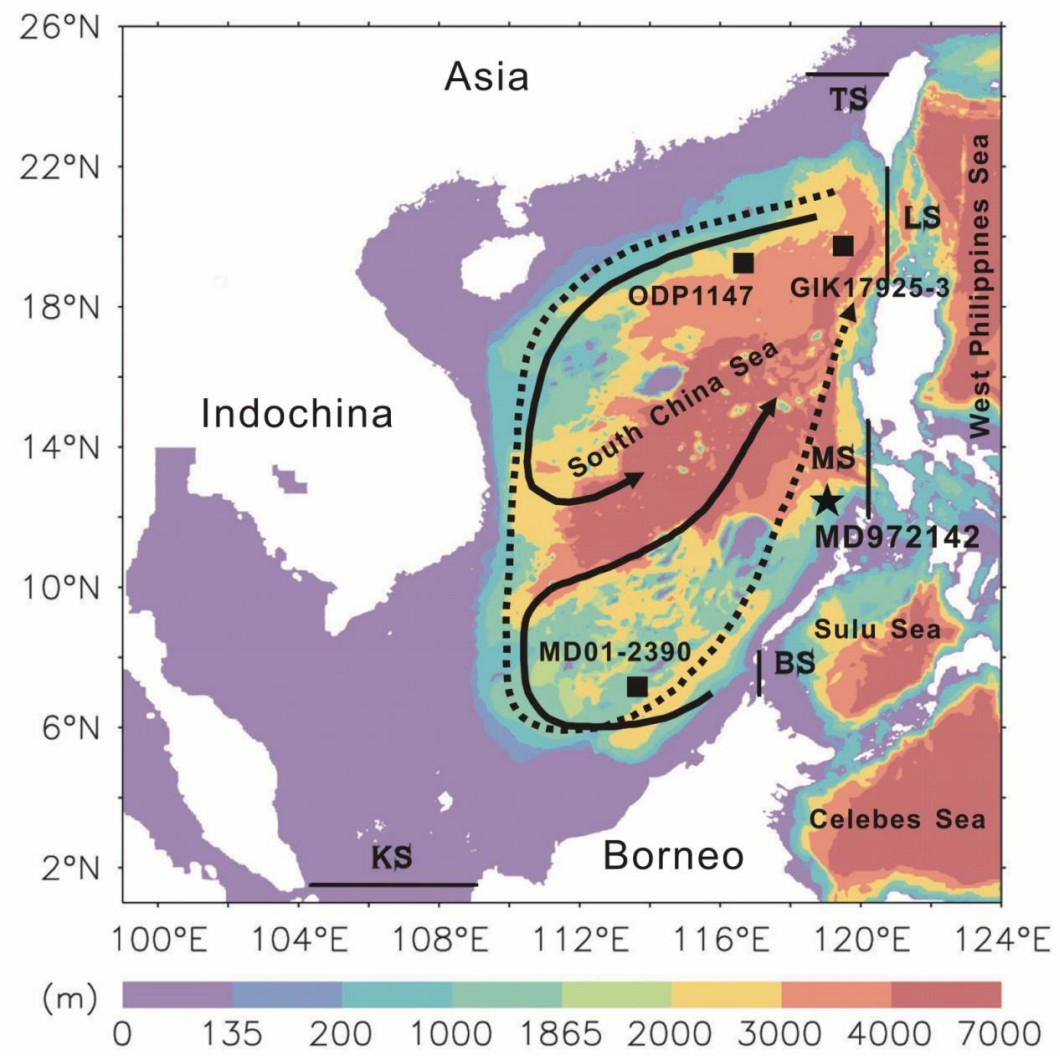

Figure 1. Bathymetry of the South China Sea and location of sediment cores referred in the text. LS, TS, KS, BS, and MS denote Luzon Strait, Taiwan Strait, Karimata Strait, Balabac Strait, and Mindoro Strait respectively. Seasonal surface circulations are drawn for winter (dashed line) and summer (solid line) respectively based on Wang and Li (2009).





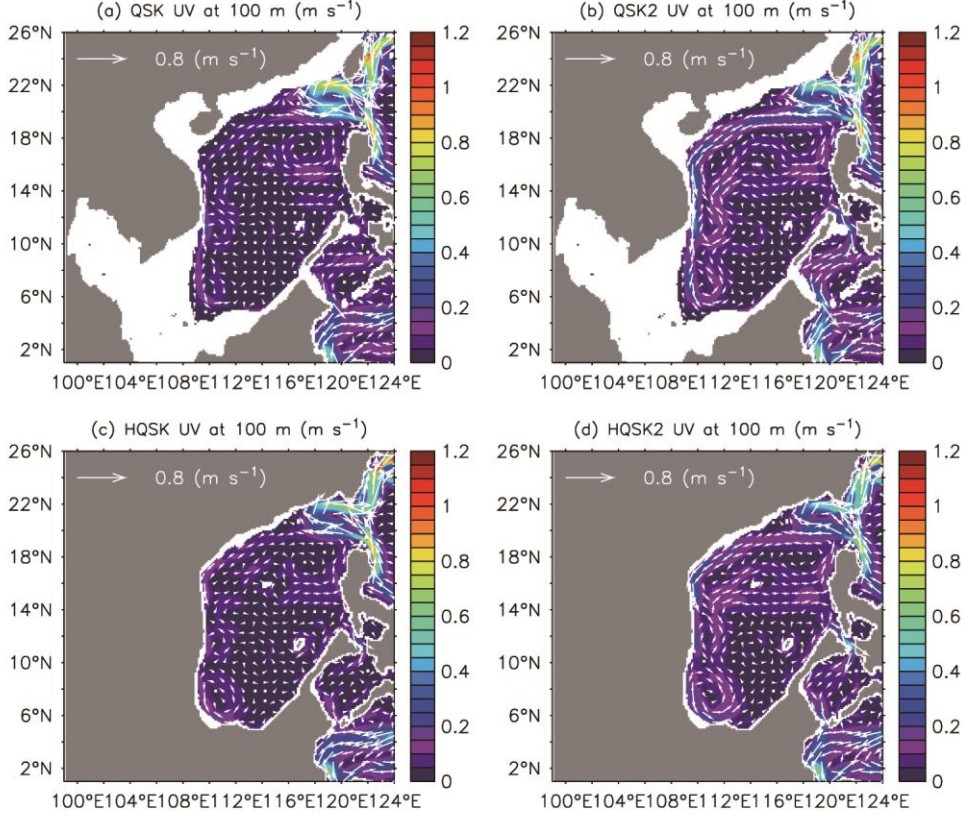

Figure 2. Flow patterns (m s$^{-1}$) for the upper 100 m in the four designed experiments. Shadings of gray are for bathymetry above sea level.





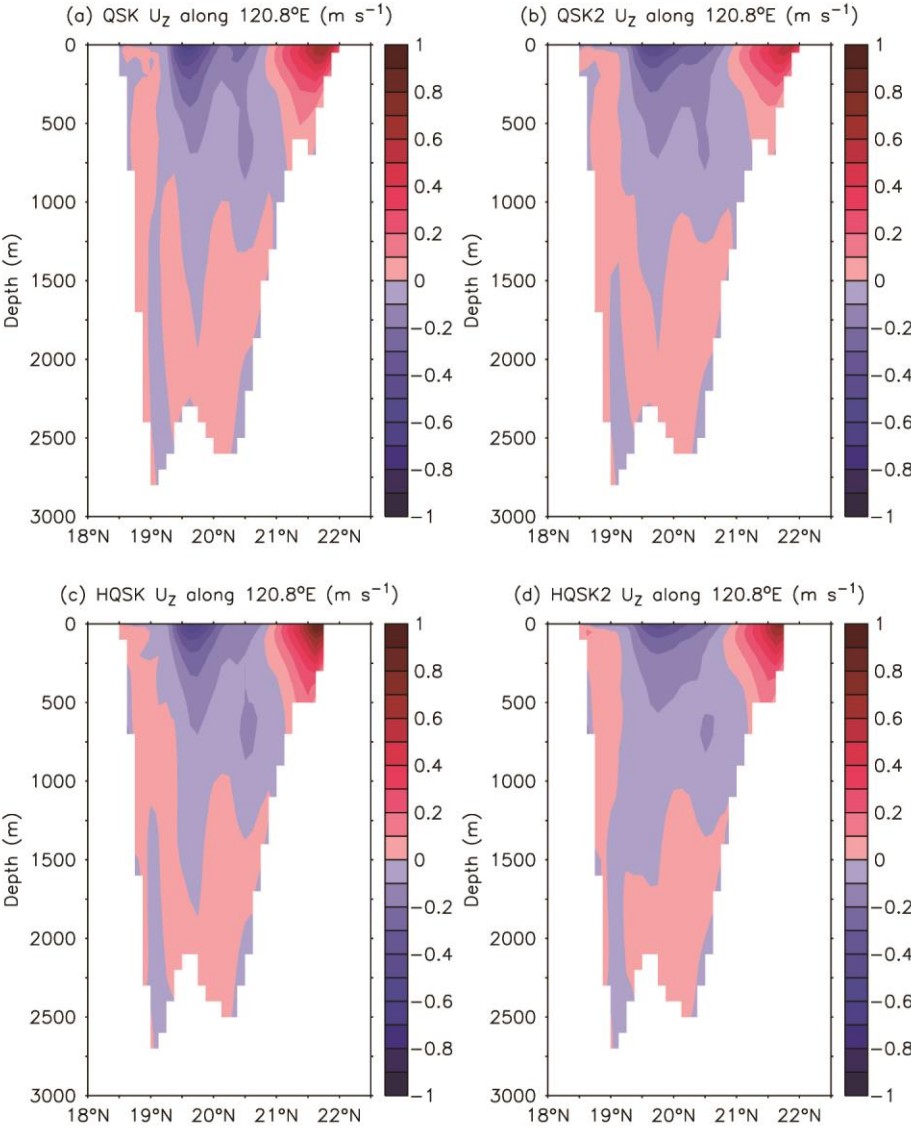

Figure 3. Zonal velocity (m s$^{-1}$) along 120.8 °E for four experiments. Positive and negative indicate eastward and westward, respectively.





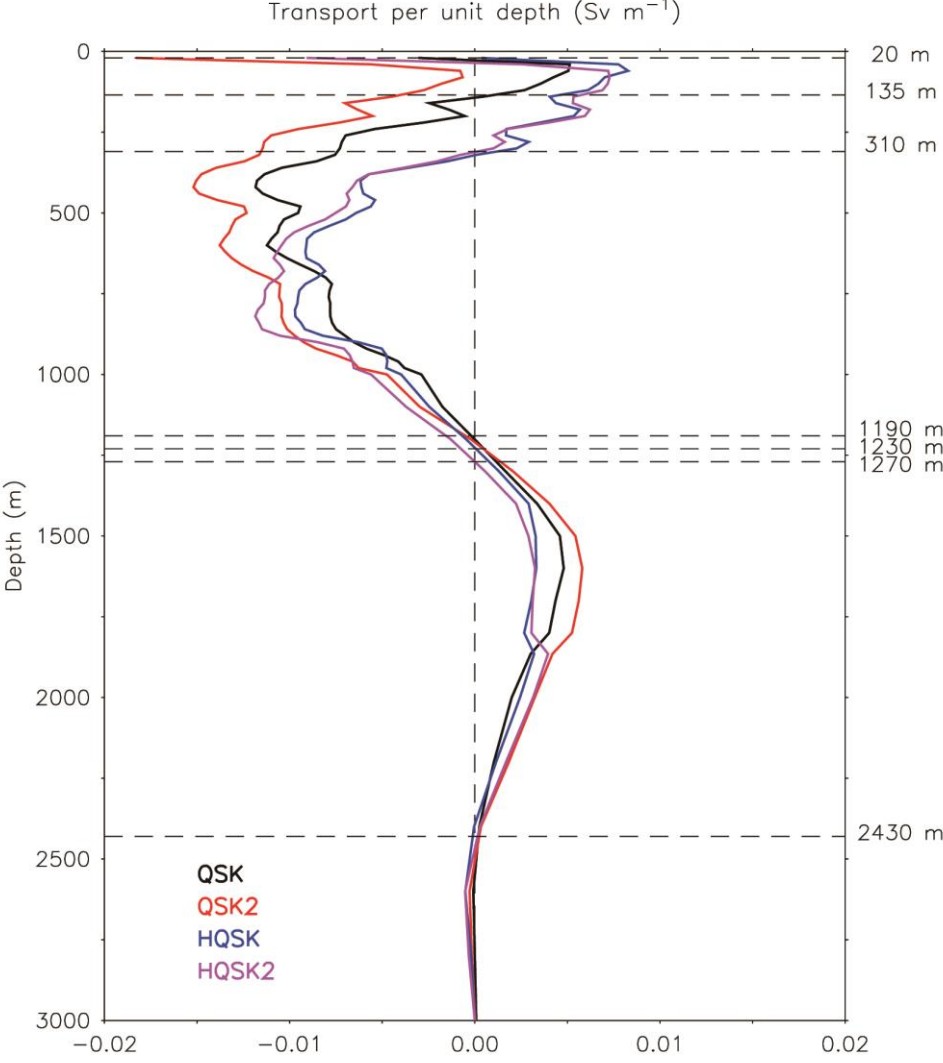

Figure 4. Annual mean vertical profiles of transport per unit depth (Sv m$^{-1}$) through the Luzon Strait for the four experiments. Positive and negative indicate eastward and westward, respectively. Dashed lines refer the depth of turning point of integrated flow in present day case (QSK in black curve).





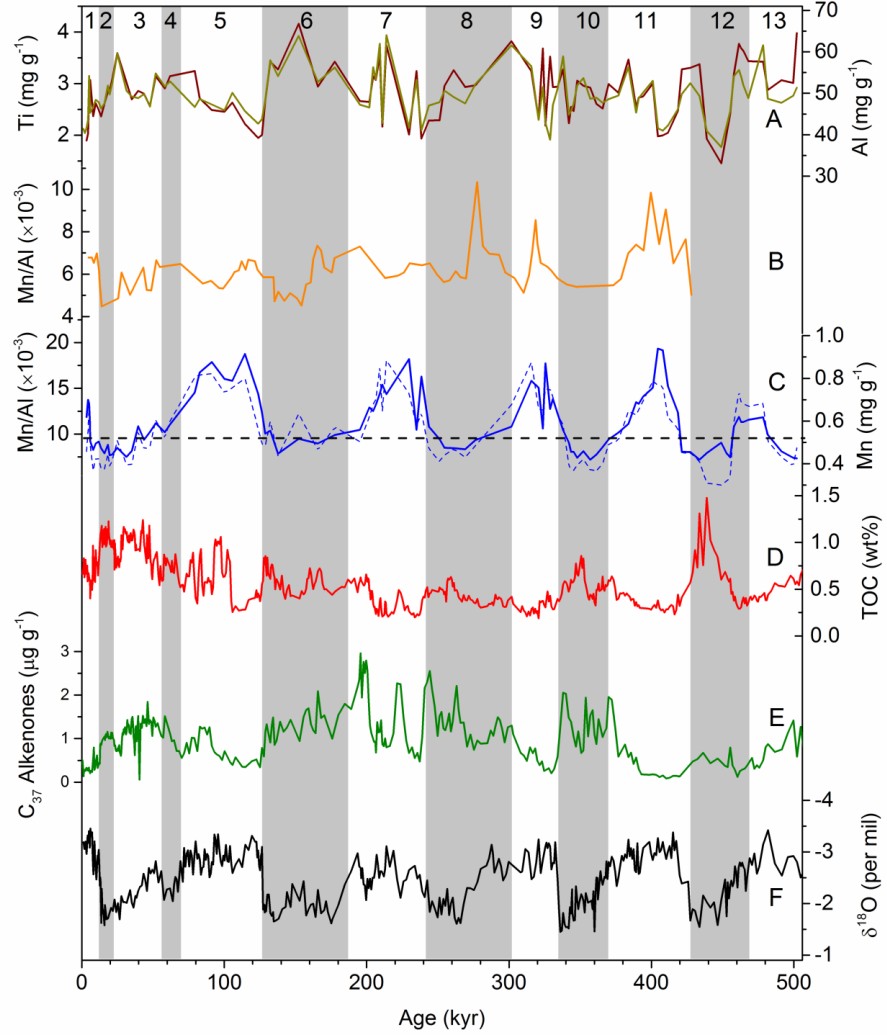

Figure 5. Paleo-records from core MD972142. (A) Content of Al (Wine, this study) and Ti (Dark yellow, this study); (B) Mn/Al ratio from core GIK17925-3 located in the northern SCS (Löwemark et al., 2009); (C) Mn/Al ratio (solid blue, this study) and content of Mn (dashed blue, this study); Black dashed line indicates average value of shale Mn/Al ratio; (D) Content of TOC (Chen et al., 2003); (E) Content of C$_{37}$ Alkenones (Shiau et al., 2008); (F) δ$^{18}$O of planktonic foraminifera *Globigerinoides ruber* (Wei et al., 2003b). Numbers on the top indicate marine isotope stages (MIS).