# Peer review of "Circulation and oxygenation of the glacial South China Sea"

_Climate of the Past, 2015_

## Referee Comment (RC1) · Anonymous Referee #1 · 26 Apr 2016

Kao et al. show a numerical simulation of the South China Sea (SCS) under both present-day and glacial sea level as well as with double wind speed. They also present a Mn/Al record from core MD972142 in the SCS. The goal of the study is to figure out whether biogeochemical changes observed at MD972142 can be explained by changes in oxygenation or changes in productivity. As the numerical study suggests that under lower sea level but stronger wind the residence time of the water in the SCS does not change much, they conclude that it is not changes in oxygenation that drove changes in biogeochemical proxies but changes in export production. The study is generally interesting and is worth publishing in Climate of the Past, if the comments below are addressed.

1) While the modeling work is interesting some links might be missing. At the moment the modeling and proxy work almost read like 2 separate stories. - There is no biogeo-

[Figure]

chemical module in the model, therefore residence time of waters in the SCS is taken as a proxy for oxygen, thus suggesting that the main O2 supply is through flushing of the SCS. Maybe a map of O2 in the SCS and of the western Pacific Ocean (water feeding the SCS) would be helpful to make that point.

- The residence time increases during low sea level from 19 to 23 years. However, if the climatological wind is doubled then the residence time under low sea level is close to present day with 18.4 years. Most of the conclusions of the paper thus rely on the hypothesis that the winds associated with both the summer and winter monsoons were twice stronger during glacial times in the SCS. Evidence for such changes should be discussed in the text. A previous study suggests that most PMIP models suggest weaker summer monsoon and about half of them suggest stronger winter monsoon.

Jiang and Lang, Last Glacial Maximum East Asian Monsoon: Results of PMIP Simulations, 2010,nDOI: http://dx.doi.org/10.1175/2010JCLI3526.1

In addition, since both winter and summer winds were strengthen, the impact of weaker summer winds during glacial time on the residence time should at least be discussed.

- It is mentioned in the abstract and in the text that the upwelling west of Luzon and east of Vietnam were enhanced. Changes in upwelling are not directly shown in the paper. A map of upwelling areas and strength for the 4 experiments could at least be shown. I guess the reason why the authors suggest the upwelling was stronger is because they assume the wind was stronger. Stronger wind leads to stronger upwelling, which leads to higher productivity. But this is entirely built on an hypothetical stronger wind. . .. So basically, the authors hypothesize that the wind was stronger during glacial times, thus leading to no significant changes in residence time. In addition, if the wind was stronger then the upwelling was stronger, which is a bit of a circular argument.

Minor comments/typos More explicit name of experiments could be useful.

P2, L.24: wrong reference to dashed line in figure 1.

Typos and grammar: Some issues with English

P 11, L10: "westward flow"

P11, L.21-22: rephrase

p12, L.20; "based on..."

p13, L. 22: "Based on ..."

P16, L.1-2: rephrase

Figure 2: Maybe a different color bar should be used (non linear). The x axis is too crowded, less numbers should be displayed.

---

## Referee Comment (RC2) · Anonymous Referee #2 · 19 May 2016

The manuscript addresses an interesting question, the glacial-time water ventilation in the South China Sea (SCS), and a 3-D modeling is used to quantify the glacial ventilation age for the first time in the region. In terms of geological data, the manuscript uses a well reported sediment core MD97-2142 (Lee, 2000; Chen et al., 2003; Wei et al. ,2003; Yu et al., 2006; Ku et al., 2008; Shiau et al., 2008; Löwemark et al 2009). Since the newly provided dataset consists only of Ti and Al curves with not-high time resolution (ca. 4 kyr), the focus of the manuscript is laid on the numerical modeling. However, the manuscript challenges some common views on the paleo-monsoon in the SCS, but failed to provide convincing arguments.

The doubled wind intensity and lowered sea level are the two backbones of the modeling experiments, but the most suspicious aspect in the modeling experiments is the hypothetical doubled wind intensity regardless which monsoon wind, winter or summer. Primary productivity in the SCS today is largely driven by the winter monsoon, and plenty of evidence indicate that the winter monsoon intensified in the SCS during the glacials resulting in enhanced productivity. By contrast, the summer monsoon was reduced during the glacial time. This notion is well supported by a variety of proxy data from marine sediments, including foraminifera (Huang et al., 1997, Marine Micropaleontology), grain size (Wang et al.,1999, Marine Geology), pollen (Sun et al., 1999,2003, Marine Geology) and isotopes (Tian et al.,2005, Paleoceanography), just to name a few. In the terrestrial realm, the extensive records of speleotheme oxygen isotope convincingly show the weakening of summer monsoon in the glacials. Therefore, the assumed doubled summer and winter monsoon intensity is in a direct conflict with the geological data.

Actually, the authors should not ignore the difference between winter and summer monsoons. Previous authors have already noticed the different upwellings between the NE and SW coasts of the SCS: intensified upwelling off Eastern Vietnam during interglacials and off the northwestern Philippines during glacial (Jian et al., 2001, Quat. Res.; Wei et al., 2006, Paleoceanography). The paleo-records are well corresponding to the modern observations with summer-monsoon inducing upwelling off Vietnam, and winter-monsoon inducing upwelling off Luzon. Surprisingly, the authors "glacial model exhibits stronger upwellings at the west off Luzon Island and the east off Vietnam" together. To the reviewer's knowledge, it is hard to imagine a mechanism in climate dynamics that could intensify both winter and summer monsoons simultaneously during the glacial time.

---

## Author Comment (AC1) · 15 Jun 2016

**Reply to Anonymous Referee #1**

Kao et al. show a numerical simulation of the South China Sea (SCS) under both present-day and glacial sea level as well as with double wind speed. They also present a Mn/Al record from core MD972142 in the SCS. The goal of the study is to figure out whether biogeochemical changes observed at MD972142 can be explained by changes in oxygenation or changes in productivity. As the numerical study suggests that under lower sea level but stronger wind the residence time of the water in the SCS does not change much, they conclude that it is not changes in oxygenation that drove changes in biogeochemical proxies but changes in export production. The study is generally interesting and is worth publishing in Climate of the Past, if the comments below are addressed.

1) While the modeling work is interesting some links might be missing. At the moment the modeling and proxy work almost read like 2 separate stories. - There is no biogeo-chemical module in the model, therefore residence time of waters in the SCS is taken as a proxy for oxygen, thus suggesting that the main $O_2$ supply is through flushing of the SCS. Maybe a map of $O_2$ in the SCS and of the western Pacific Ocean (water feeding the SCS) would be helpful to make that point.

**Reply:** Thanks for this constructive suggestion. We will add a dissolved $O_2$ profile into Figure 1 and add more illustrations why the through flow of the intermediate water is important in determining the basin wide redox status. This will be a link to connect modeling work and biogeochemical records as indicated by reviewer.

The biogeochemical module will be added in our future works; however, it is a long way to go to properly validate the three dimensional biogeochemistry (i.e., not only the surface Chl-a but also the vertical oxygen and POM distribution). At current stage, we used the physical model that had been very well validated by modern physical observations in the SCS (Hsin et al., 2008, 2010, 2012).

[Figure]

$O_2$ profile along the red band on insert map. $O_2$ data source from WOA2013, http://www.nodc.noaa.gov/OC5/woa13/woa13data.html.

- The residence time increases during low sea level from 19 to 23 years. However, if the climatological wind is doubled then the residence time under low sea level is close to present day with 18.4 years. Most of the conclusions of the paper thus rely on the hypothesis that the winds

associated with both the summer and winter monsoons were twice stronger during glacial times in the SCS. Evidence for such changes should be discussed in the text. A previous study suggests that most PMIP models suggest weaker summer monsoon and about half of them suggest stronger winter monsoon. Jiang and Lang, Last Glacial Maximum East Asian Monsoon: Results of PMIP Simulations, 2010, DOI: http://dx.doi.org/10.1175/2010JCLI3526.1 In addition, since both winter and summer winds were strengthen, the impact of weaker summer winds during glacial time on the residence time should at least be discussed.

**Reply:** Basing on this comment, we have carried out two new cases for the low sea level condition. In the first new case (HQSK2b), summer wind was reduced 50% and winter wind remained unintensified, i.e. modern winter wind intensity. From this case, we realized the intermediate water through flow reduced to 4.69 Sv resulting in a slightly longer residence time of 26.3 year (see Table 1 below). The second case (HQSK2a) was set for stronger winter monsoon (1.2 times of modern winter wind) and reduced summer monsoon (0.75 times of modern summer wind) according to the result from PMIP published by Jiang and Lang (2010). The intermediate water through flow was 4.95 Sv, which is comparable but resulting in a slightly shorter residence time of 24.9 year. Both new cases gave slightly longer residence times as speculated by reviewer, however, the oxygenation state deduced from these new results would not alter our original story. In this revised version, we will also add more illustrations for the reasons of wind speed selection. The mentioned reference will be added also.

Table 1. Estimated residence time for the SCS based on the net transport of intermediate water through the Luzon Strait.

| Case lable | QSK | QSK2 | HQSK | HQSK2 | HQSK2a | HQSK2a |
|---|---|---|---|---|---|---|
| LS TP interval (m) | 135–1190 | 20–1190 | 310–1230 | 310–1270 | 310–1230 | 310–1230 |
| TP (Sv) | -7.04 | -10.41 | -5.35 | -6.71 | -4.95 | -4.69 |
| RT (year) | 19.0 | 12.8 | 23.0 | 18.4 | 24.9 | 26.3 |

- It is mentioned in the abstract and in the text that the upwelling west of Luzon and east of Vietnam were enhanced. Changes in upwelling are not directly shown in the paper. A map of upwelling areas and strength for the 4 experiments could at least be shown. I guess the reason why the authors suggest the upwelling was stronger is because they assume the wind was stronger. Stronger wind leads to stronger upwelling, which leads to higher productivity. But this is entirely built on an hypothetical stronger wind……So basically, the authors hypothesize that the wind was stronger during glacial times, thus leading to no significant changes in residence time. In addition, if the wind was stronger then the upwelling was stronger, which is a bit of a circular argument.

**Reply:** In fact, changes in upwelling west of Luzon and east of Vietnam for these 4 cases were displayed in Figure 2 in old version. However, the color bar we applied did not properly reveal these differences. We will redraw Figure 2 using non-linear style color bar. As replied above, we ran two more cases with reduction summer wind, which provided consistent results, to support our hypothesis.

Minor comments/typos

More explicit name of experiments could be useful.

**Reply:** More explicit names will be provided in the following revised version. .

P2, L.24: wrong reference to dashed line in figure 1.

**Reply:** Corrected. Summer surface circulation is represented by solid line in Figure 1.

Typos and grammar: Some issues with English

P 11, L10: "westward flow"

**Reply:** Corrected. It is westward flow.

P11, L.21-22: rephrase

**Reply:** We will add results from these two new cases and revise this the following revised version.

p12, L.20; "based on..."

**Reply:** Corrected.

p13, L. 22: "Based on ..."

**Reply:** Corrected.

P16, L.1-2: rephrase

**Reply:** We made revision as follows:

"The content of Al is often used to infer the terrestrial input (Brumsack, 2006) and the normalization onto Al may eliminate the dilution effect on proxies. However, recent studies have indicated that sedimentary Al may have biogenic source (Wei, G. et al., 2003; Murray and Leinen, 1996). Thus, before applying the Mn/Al ratio for discussing environmental redox change, we need to confirm sedimentary Al in our core was not influenced by biogenic process."

Figure 2: Maybe a different color bar should be used (non linear). The x axis is too crowded, less numbers should be displayed.

**Reply:** We will redraw Figure 2 by using proper color bar.

---

## Author Comment (AC2) · 15 Jun 2016

**Reply to Anonymous Referee #2**

The manuscript addresses an interesting question, the glacial-time water ventilation in the South China Sea (SCS), and a 3-D modeling is used to quantify the glacial ventilation age for the first time in the region. In terms of geological data, the manuscript uses a well reported sediment core MD97-2142 (Lee, 2000; Chen et al., 2003; Wei et al. ,2003; Yu et al., 2006; Ku et al., 2008; Shiau et al., 2008; Löwemark et al 2009). Since the newly provided dataset consists only of Ti and Al curves with not-high time resolution (ca. 4 kyr), the focus of the manuscript is laid on the numerical modeling. However, the manuscript challenges some common views on the paleo-monsoon in the SCS, but failed to provide convincing arguments.

   The doubled wind intensity and lowered sea level are the two backbones of the modeling experiments, but the most suspicious aspect in the modeling experiments is the hypothetical doubled wind intensity regardless which monsoon wind, winter or summer. Primary productivity in the SCS today is largely driven by the winter monsoon, and plenty of evidence indicate that the winter monsoon intensified in the SCS during the glacials resulting in enhanced productivity. By contrast, the summer monsoon was reduced during the glacial time. This notion is well supported by a variety of proxy data from marine sediments, including foraminifera (Huang et al., 1997, Marine Micropaleontology), grain size (Wang et al.,1999, Marine Geology), pollen (Sun et al., 1999, 2003, Marine Geology) and isotopes (Tian et al., 2005, Paleoceanography), just to name a few. In the terrestrial realm, the extensive records of speleotheme oxygen isotope convincingly show the weakening of summer monsoon in the glacials. Therefore, the assumed doubled summer and winter monsoon intensity is in a direct conflict with the geological data.

**Reply:** This question was also raised by Reviewer #1. As replied to Reviewer #1, we have carried out two new cases with reducing summer monsoon wind. The oxygenation state deduced from the new model runs would not alter our story at all. In the following revised version, we will add results from these two cases.

   Actually, the authors should not ignore the difference between winter and summer monsoons. Previous authors have already noticed the different upwellings between the NE and SW coasts of the SCS: intensified upwelling off Eastern Vietnam during interglacials and off the northwestern Philippines during glacial (Jian et al., 2001, Quat. Res.; Wei et al., 2006, Paleoceanography). The paleo-records are well corresponding to the modern observations with summer-monsoon inducing upwelling off Vietnam, and winter-monsoon inducing upwelling off Luzon. Surprisingly, the authors "glacial model exhibits stronger upwellings at the west off Luzon Island and the east off Vietnam" together. To the reviewer's knowledge, it is hard to imagine a mechanism in climate dynamics that could intensify both winter and summer monsoons simultaneously during the glacial time.

**Reply:** Reviewer is correct about the upwelling center distribution in seasonal scale. However, the contour maps presented in Figure 2 are the annual average; thus, both

upwelling centers can be seen. The new model cases (with reducing summer wind) requested by reviewer indeed show weakened upwelling to the east of Vietnam as speculated by reviewer. The results of new cases will be added into old Figure 2. More descriptions will be made to clarify the differences in circulation pattern in annual basis.

Also, we will present the profile of monthly flow through the Luzon Strait to reveal the seasonality in water exchange.